# Itaconate Isomers in Bread

**DOI:** 10.3390/antiox11071382

**Published:** 2022-07-16

**Authors:** Mona Gruenwald, Fangfang Chen, Heike Bähre, Frank Pessler

**Affiliations:** 1M.Sc. Program in Food Research and Development, Leibniz University Hannover, 30167 Hannover, Germany; mona.gruenwald@web.de; 2Research Group Biomarkers for Infectious Diseases, TWINCORE Centre for Experimental and Clinical Infection Research, 30625 Hannover, Germany; fangfang.chen@twincore.de; 3Research Group Biomarkers for Infectious Diseases, Helmholtz Centre for Infection Research, 38124 Braunschweig, Germany; 4Research Core Unit Metabolomics, Hannover Medical School, 30625 Hannover, Germany; baehre.heike@mh-hannover.de

**Keywords:** antioxidants, baking, citraconic acid, dicarboxylic acids, diet, food, itaconic acid, mesaconic acid, nutrients, nutrition

## Abstract

The naturally occurring isomers itaconate, mesaconate and citraconate possess immunomodulatory, antioxidative and antimicrobial properties. However, it is not known whether they occur in commonly consumed human foods. Considering that they can arise as a result of heat conversion, we tested whether they occur in bread, representing a commonly consumed baked good. Using high-performance liquid chromatography–tandem mass spectrometry, we measured concentrations of the three isomers and their potential precursors, citrate and *cis*-aconitate, in unbaked sourdough and dough, and in crumb and crust of baked bread. All three isomers were detected at low concentrations (<20 pmol/mg dry weight) in sourdough, dough, crumb and crust. Concentrations of itaconate and citraconate were substantially higher in crust than in crumb of wheat and rye bread, and a modest increase in mesaconate was observed in crust of rye bread. In contrast, *cis*-aconitate concentrations were considerably lower in crust, which was consistent with the conversion of *cis*-aconitate to itaconate isomers due to higher temperature of the dough surface during baking. Based on data on the average consumption of bread and related baked goods in Germany, the daily intake of itaconate isomers was estimated to be roughly 7–20 µg. Thus, baked goods constitute a regular dietary source of low amounts of itaconate isomers. In order to enable studies on the impact of dietary intake of itaconate isomers on human health, their concentrations should be assessed in other foods that are subjected to high heating.

## 1. Introduction

In recent years, a multitude of publications have highlighted immunomodulatory, cytoprotective, antibacterial and antiviral properties of itaconic acid [1]. This small dicarboxylic acid is synthetized de novo during macrophage activation via decarboxylation of the tricarboxylic acid (TCA) cycle intermediate *cis*-aconitate [2]. Loss-of-function mutations in the gene encoding the corresponding enzyme, *cis*-aconitate decarboxylase (ACOD1, formerly known as immune-responsive gene 1, IRG1) are very rare, suggesting that physiologic synthesis of itaconate has been important in human evolution [3]. Citraconate and mesaconate are isomers of itaconate that differ only by the location of the internal double bond (Figure 1), suggesting that they may have similar properties. The pathways leading to the synthesis of mesaconate and citraconate have been partially elucidated in mice and/or humans. One source of mesaconate is its catabolism from itaconate during macrophage activation [4,5], whereas citraconate may arise from isoleucine catabolism [6]. In addition, there are isolated reports of their detection in bacteria [7,8,9], but little is known about how common or biologically relevant the respective pathways are in prokaryotes. We have recently shown that all three isomers occur in the organs of healthy mice and in human blood, although the distribution in different blood fractions varies [10]. Notably, He et al. [4] and we [5] recently found that exogenously added mesaconate exerts immunomodulatory effects similar to itaconate. We also reported that all three isomers exert antioxidative, immunomodulatory and antiviral effects, and that these effects are relatively independent of the target cell type [11,12]. Notably, citraconate is the most potent agonist of the cytoprotective and anti-inflammatory NRF2 pathway among the three isomers [5]. Despite this accumulating evidence that all three isomers may have beneficial effects on human cells, essentially nothing is known about their occurrence in food. We have, therefore, tested whether any of them occur in bread. We selected bread for three reasons: (i) it is a nearly universally consumed staple, (ii) the precursors of itaconic acid, *cis*- and *trans*-aconitic acid, have been detected in wheat seedlings [13] and (iii) itaconic, mesaconic and citraconic acids can originate from citric or *cis*-aconitic acid via heat conversion [14,15]. Indeed, we detected low concentrations of all three isomers in bread and found that concentrations are substantially higher in crust, likely due to higher heat exposure during baking.

## 2. Materials and Methods

### 2.1. Homemade Bread

To make type I sourdough (as defined by [16]), wheat whole-grain flour (Weizenvollkornmehl, EDEKA Bio) or rye whole-grain flour (Roggenvollkornmehl, Alnatura) was mixed with an equal amount of water in a clean container and kept at room temperature (RT). After 24 h, part of the flour–water mixture was mixed with fresh flour and water. This backslopping process was repeated until the volume of the mixture had doubled within 24 h. Before adding the sourdough to the other ingredients, one tablespoon (2 g) was taken as a defined sample and stored at −20 °C. To make the dough, the sourdough was mixed with equal volumes of water (35 °C) and whole-grain flour and allowed to rise for 4 h at RT. Then, 80 g sourdough was mixed with 120 mL water (12 °C) and 200 g whole-grain flour (11 °C) and allowed to rise for 1 h at a dough temperature of 21 °C. Next, 20 mL of water (12 °C) was mixed with the dough. After 10 minutes (min) of mixing, 4 g of salt was added and gently blended. The dough was left to rise for 6 h (dough temperature 23 °C). It was stretched and folded after 30, 60 and 90 min. After doubling in volume (approx. 6 h), the dough was shaped and allowed to rise in a proofing basket for 10–14 h. Then, 5 g of dough was taken as a defined sample just before baking and stored at −20 °C. The bread was baked for 15 min at 250 °C with steam and then another 20 min at 220 °C without steam. This process was repeated from scratch on three different days, resulting in three loaves of homemade wheat or rye bread, each weighing about 500 g, for analysis.

### 2.2. Commercial Bread

Three loaves each of mixed wheat bread (1688 Das Milde, 250 g loaf) and mixed rye bread (1688 Steinofenbrot, 250 g loaf; both Harry-Brot GmbH, Schenefeld, Germany) were bought in a local supermarket. The following ingredients were listed on the packaging of the wheat bread: wheat flour, water, natural sourdough (water, whole-grain rye flour, rye flour), rye flour, salt, yeast, rapeseed oil, inverted sugar syrup, emulsifier mono- and diglycerides of fatty acids and acidity regulator sodium acetate. The following ingredients were listed for the rye bread: wheat flour, natural sourdough (water, whole-grain rye flour, rye flour), water, rye flour, salt, yeast, rye malt flour and acidity regulator sodium acetate.

### 2.3. Preanalytical Processing

Crumb samples were obtained from the center of a slice of bread. Crust samples were taken to a depth of 0.5 cm from the bread surface. Sourdough and dough samples had been stored at −20 °C and were thawed for further processing. From all samples (sourdough, dough, crumb and crust), 100 mg was transferred into a 2 mL FastPrep tube and mixed with 100 µL of extraction solvent (acetonitrile/methanol/water, 2/2/1, *v/v/v*) containing an internal standard (ISTD; 1 µM ^13^C_2_-Citrate, 2 µM ^13^C_6_-*cis*-aconitate, 1 µM ^13^C_5_-itaconate). Samples were standardized for lyophilization with the addition of 400 µL water and then frozen at −80 °C. Samples were lyophilized directly in the 2 mL FastPrep tubes, which were covered with a thin paper tissue to prevent loss of solid components during the process. Lyophilization was carried out for 48 h in an Alpha 2–4 LSC Basic lyophilizer (Martin Christ Gefriertrocknungsanlagen GmbH, Osterode, Germany) equipped with a Vacuum-Pure 10 C pump (Vacuubrand, Wertheim, Germany) at a pressure of 0.08 mbar. After lyophilization, lysing matrix A (6910, MP Biomedicals, Santa Ana, CA, USA) was added to the FastPrep tubes. Then, 200 µL of water and 800 µL of ice-cold extraction solvent (acetonitrile/methanol, 1/1, *v/v*) were added to the samples. The samples were homogenized at RT using a FastPrep^®^-24 Instrument (MP Biomedicals, Santa Ana, CA, USA) at a speed of 4.0 m/s for 2 × 30 s. The subsequent extraction steps were performed as described in [10].

### 2.4. Wheat Seedlings

Commercially available wheat seeds (*Triticum aestivum*, Sperli; obtained from Sperli GmbH, Everswinkel, Germany) were germinated and grown under natural light for 7 days until the first foliage leaf became visible. Cotyledons and foliage leaves from several plants were harvested, cut into 0.5 × 0.5 cm squares and stored at −20 °C until further processing. Frozen samples were transferred into 2 mL FastPrep tubes filled with lysing matrix A. They were mixed with 200 µL of water and 800 µL of ice-cold extraction solvent, and 100 µL of extraction solvent with ISTD. The samples were homogenized at RT using a FastPrep^®^-24 Instrument at a speed of 4.0 m/s for 2 × 30 s. The subsequent extraction steps were performed as described in [10].

### 2.5. Mass Spectrometry

Concentrations of citraconate, mesaconate, itaconate, citrate and *cis*-aconitate were measured by high-performance liquid chromatography–tandem mass spectrometry (HPLC-MS/MS). As described in [4], this process used a Nexera chromatography system consisting of a controller (CBM-20A), an autosampler (SIL-30AC), two pumps (LC-30AD), a degasser (DGU-20A5) and a column oven (CTO-20AC, Shimadzu, Japan), coupled with a QTRAP5500 triple quadrupole/linear ion trap mass spectrometer (Sciex, Framingham, MA, USA). Data acquisition and further quantification were performed using Analyst^®^ Software 1.7 (Sciex, Framingham, MA, USA).

### 2.6. Computation and Statistics

The significance of differences between groups was assessed by two-way ANOVA with Tukey multiple comparison tests, where statistical significance was defined as *p* ≤ 0.05. The estimated daily intake of itaconate isomers in bread and related baked products was calculated in two ways. According to the European Prospective Investigation into Cancer and Nutrition (EPIC)-Potsdam study [17], the average consumption of whole-grain bread per person per day in Germany is 45.3 g. Based on this, the estimated daily intake of the itaconate isomers in whole-grain bread was calculated using the isomer concentrations measured in homemade bread. The information on bread consumption in Germany from the National Nutrition Survey II [18] considers bread as part of a food group which, in addition, includes rolls, baguettes, toast and rusks. The daily consumption of this group by women (111 g) and men (158 g) averaged 134.5 g. Based on this, the estimated daily intake of the itaconate isomers from the expanded bread/baked goods group was calculated using the values measured in both homemade and commercial bread.

## 3. Results

### 3.1. Itaconate Isomers in Homemade and Commercial Bread

All three isomers were readily detected in the crumb and crust of common commercially available brands of wheat and rye bread. Concentrations of itaconate and citraconate were significantly higher in crust than in crumb, whereas differences in mesaconate were much smaller and not significant (Figure 2A–C). Citrate concentrations were also significantly higher in crust, whereas *cis*-aconitate concentrations were higher in crumb (Figure 2D, E). Itaconic and citraconic acids can be produced by heating citric acid, whereby itaconic acid additionally arises upon the heating of *cis*-aconitic acid [14,15] and mesaconic acid is a by-product of heating citraconic acid [19]. To test whether any of these isomers accumulated during baking, we measured their concentrations in homemade wheat and rye bread, comparing sourdough starter, unbaked dough, crumb and crust. Low concentrations of all isomers were detected in sourdough, dough and crumb, and significantly higher concentrations of itaconate and citraconate in crust (Figure 2F–G). Mesaconate differed in that only a mild increase was detected in the crust of wheat bread, but higher concentrations in the dough of rye bread, which did not increase further in crust (Figure 2H). Citrate gave similar results to mesaconate, which were not significant (Figure 2I). Of note, *cis*-aconitate concentrations decreased significantly in wheat and rye bread, from sourdough (highest) to crust (lowest) (Figure 2J).

We then tested whether the molar decrease in *cis*-aconitate from crumb to crust accounted for the molar gain in itaconate isomers. With the exception of commercial rye bread, the gain of itaconate isomers in crust was several moles larger than the loss of *cis*-aconitate, suggesting the presence of other heat-labile precursors (Figure 3).

Next, we calculated the amount of each isomer in a slice of bread. We first determined the relative amount of crumb and crust in a slice of commercial bread: in a slice weighing 56 g, 54 g corresponded to crumb and 2 g to crust (27:1). The calculated analyte concentrations were normalized using the wet weights before lyophilizing the samples, as lyophilization reduces the weight of the crumb by 35–51% and the crust by 10–32%. The resulting calculated isomer amounts in commercial wheat or rye bread are summarized in Table 1.

### 3.2. Estimated Dietary Intake of Itaconate Isomers in Bread

To explore whether the consumption of bread could lead to the ingestion of biologically meaningful amounts of any of the isomers, we estimated their intake based on two studies of bread consumption in Germany. According to the EPIC–Potsdam study, the average daily consumption of whole-grain bread in Germany is 45.3 g per day [17]. The German National Nutrition Survey II reports a daily intake of bread and related baked goods of 134.5 g [18]. Using these two estimates, we calculated the approximate dietary intake of the isomers based on the measured amounts of the isomers in bread shown in Table 1. The results are shown in Table 2.

### 3.3. Absence of Itaconate Isomers in Wheat Seedlings

Motivated by a report that wheat seedlings contain the precursors of itaconate, *cis*- and *trans*-aconitate [13], the LC-MS/MS assay was also applied to wheat seedlings. Substantial concentrations of citrate and *cis*-aconitate were detected, which were much higher than in the bread-related samples (Table 3). However, concentrations of the three itaconate isomers were below the limit of detection of the assay.

## 4. Discussion

In this first analysis of itaconate isomers in common human food, we found that all three isomers occur in wheat and rye bread, whereby the substantially higher concentrations of itaconate and citraconate in crust likely result from heat conversion from *cis*-aconitate during baking. However, in general, the molar increase in itaconate isomers was greater than the decrease in *cis*-aconitate, suggesting the presence of additional precursors. As an intermediate of the tricarboxylic acid cycle, *cis*-aconitate likely occurs in all human foods. In order to get a more complete view of the dietary intake of itaconate isomers, it is now important to measure their concentrations in other foods that are subjected to high heating, such as fried, deep-fried, roasted and grilled foods. Some variation in isomer concentrations between commercial and homemade or wheat and rye bread may have been due to differences in fermentation products, as microbial populations were likely not identical in the sourdough used for each of the four bread types, and only the commercial bread contained yeast as a leavening agent.

Considering the lack of data on itaconate isomers in human organs, it is difficult to judge the biological relevance of the measured values. A study from the 1950s on itaconate supplementation of a rat diet demonstrated that itaconate can be absorbed orally and that supplementation of up to 2% did not lead to histological changes in internal organs, but 1% supplementation significantly reduced weight gain, presumably by inhibiting succinate dehydrogenase activity [20]. However, it was not reported whether the diminished weight gain was due to a general growth inhibition or a preferential effect on fat accumulation. The potential impact of itaconate on organismal redox balance was recently addressed in a study on itaconate supplementation of chicken feed [21]. These authors showed that adding up to 1% of itaconate to feed had several desirable effects of economic importance, including higher breast and thigh muscle yield, but also led to positive changes in objective biomarkers of amino acid synthesis and antioxidative responses. However, the amounts of itaconate ingested by the animals in both studies were 5–6 orders of magnitude higher than the estimated human intake from bread: assuming an energy content of organic acids of 3 kcal/g and an average caloric intake of 2300 kcal per day, isomer intake from bread would correspond to approximately 21–57 × 10^−6^ kcal/day, i.e., 1/10^−8^–2.5/10^−8^ (0.000001–0.0000025%) of daily caloric intake. Considering the potential antibacterial properties of itaconate, one may postulate that the relatively low dietary amounts of itaconate isomers could modulate intestinal microflora, but there are no data yet to substantiate this hypothesis. Even though there are currently no studies about the impact of increasing enteral itaconate isomer intake on human health, the available evidence from cellular and preclinical models suggests that the net effect of relatively high concentrations might be positive. This is highlighted in two recent reviews on potential applications of itaconate for the treatment of cardiovascular diseases [22,23]. Assuming that it will not be possible to increase dietary intake substantially through other common foods, the development of nutritional supplements based on itaconate isomers may be a feasible method.

## 5. Conclusions

In summary, the present study should be considered as proof of concept that low amounts of itaconate isomers occur in bread and related baked goods, whereby heat conversion from *cis*-aconitate and other (not identified) precursors likely explains the higher concentrations found in crust than in crumb. These results should, therefore, stimulate further investigations into their occurrence in other foods that are subjected to high heating during preparation.

## Figures and Tables

**Figure 1 antioxidants-11-01382-f001:**
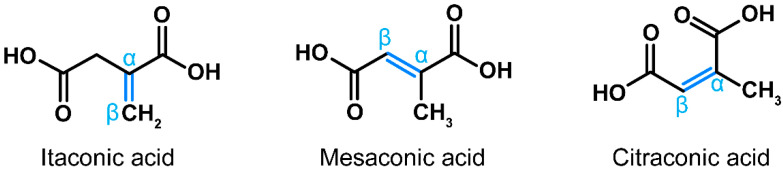
Structures of itaconic, mesaconic and citraconic acids. The isomers differ only by the arrangement of the internal double bond (marked blue), which is an important determinant of electrophilicity and antioxidative potential.

**Figure 2 antioxidants-11-01382-f002:**
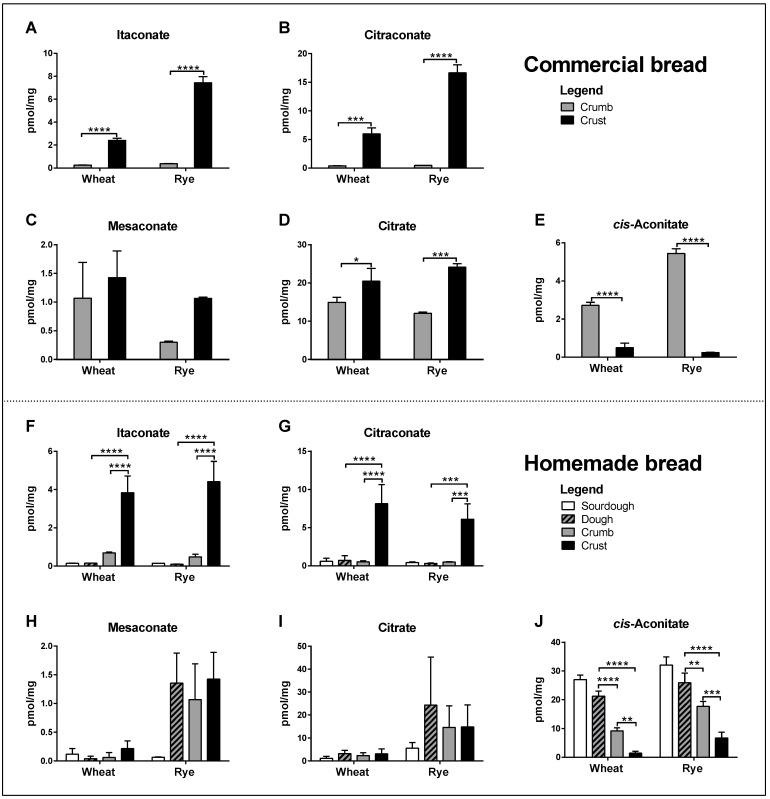
Comparison of analyte concentrations (pmol/mg dry weight) in crumb and crust of commercial wheat and rye bread (**A**–**E**) and in sourdough, dough, crumb and crust of homemade whole-grain wheat and rye bread (**F**–**J**). The data are a pool of three independent experiments, each comprising 3 non-technical replicates (individual sample aliquots) per sample, i.e., 9 data points per bar. Two-way ANOVA with Tukey test for multiple testing. * *p* ≤ 0.05, ** *p* ≤ 0.01, *** *p* ≤ 0.001, **** *p* ≤ 0.0001.

**Figure 3 antioxidants-11-01382-f003:**
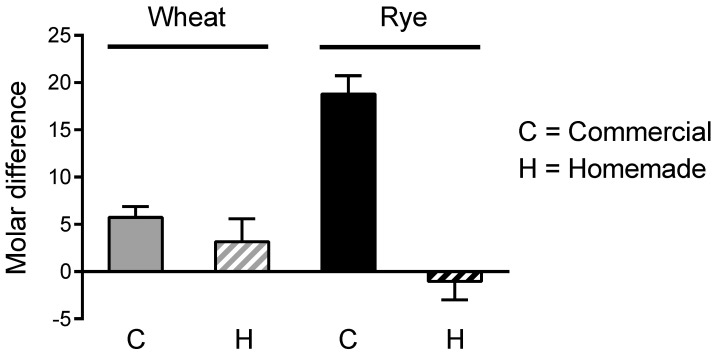
Crust vs. crumb: comparison of moles of itaconate isomers in crust gained vs. moles of *cis*-aconitate in crust lost. The difference in *cis*-aconitate concentration (moles/mg dry weight) between crust and crumb was subtracted from the difference in concentration of itaconate isomers (sum of the 3 isomers) between crust and crumb. A positive value indicates that the gain in itaconate isomers concentration is larger than the loss of *cis*-aconitate concentration. Based on the same data underlying Figure 2.

**Table 1 antioxidants-11-01382-t001:** Comparison of calculated amounts of all analytes per slice of bread (g × 10^−6^ ± SD).

Bread Type		Itaconate	Citraconate	Mesaconate	Citrate	*cis*-Aconitate
Wheat	Commercial	1.5 ± 0.1	2.7 ± 0.1	0.8 ± 0.04	96.0 ± 6.6	15.0 ± 0.2
	Homemade	3.4 ± 0.3	3.7 ± 1.0	0.3 ± 0.4	14.0 ± 7.2	46.4 ± 4.3
Rye	Commercial	2.8 ± 0.2	4.8 ± 0.3	1.3 ± 0.1	72.0 ± 4.0	26.7 ± 0.6
	Homemade	2.7 ± 0.6	3.1 ± 0.5	4.1 ± 2.4	83.0 ± 53.6	86.9 ± 8.5

**Table 2 antioxidants-11-01382-t002:** Estimated daily intake of itaconate isomers in bread.

		Estimated Mean Intake (Adult/Day) g × 10^−6^			
Reference	Average Bread Intake (Adult/Day) ^c^	Itaconate	Citraconate	Mesaconate	All
[17] ^a^	45.3	2.1	2.8	1.8	7.0
[18] ^b^	134.5 ^c^	6.2	8.6	3.9	19

^a^ EPIC Potsdam: whole-grain bread. ^b^ German National Nutrition Survey II: bread and related baked goods. **^c^** Mean of intake by men and women (in g).

**Table 3 antioxidants-11-01382-t003:** Citrate and *cis*-aconitate concentrations in wheat seedlings (pmol/mg wet weight) ^a^.

	Citrate	*cis*-Aconitate
Seed leaves	477 ± 187	189 ± 21
Mature leaves	202 ± 61	218 ± 7

^a^*n* = 3 samples of each leaf type. Mean ± SD.

## Data Availability

The data presented in this study are available in the article.

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
