# Peer review of "Itaconate Isomers in Bread"

_antioxidants, 2022, doi:10.3390/antiox11071382_

Round 1

Reviewer 1 Report

This is a well written paper that starts to evaluate the relevance of a series of compounds that may occur in human foods and may have been relevant bioactivity.

I believe the paper demonstrates assessment of the general levels of the compounds, in one of the mostly eaten categories of foods. At the measured levels, intake of the itaconates would be considered low and the paper concludes that there is currently no evidence of positive or negative implications of ingesting these quantities of materials.  What makes the paper useful was the speculation of how the few analytical results obtained could be interpreted in terms of the average human diet.

I consider that there is enough data to support the authors deliberations and conclusions. The authors state that they achieved a “proof- of- concept” and I can accept the paper as such rather than a paper where the robustness of the assessments and the calculation of the factors is painstaking.

I am not an expert in the area of itaconate compounds and their assessment and therefore there would have been additions/changes to this paper that I would have found very helpful.

1.    I would have appreciated a figure showing the chemical compounds and how the different isomers relate to each other and known factors that drive the different forms.

2.    I would have appreciated knowing a bit more about the samples and their replication. I think one recipe for wheat and one for rye was used for the homemade breads and this was repeated from scratch on three occasions. But not sure this is true.

3.    For the commercial breads, were three loaves of each type bought on one occasion (i.e. one batch).

4.    Some table of the bread details would be useful, some of the data is in the paper, but hard to pull out. Details of ingredients on the commercial breads could be given (were there redox agents in the formulation?). Size of loaf, weight and volume and the size of slice would help define the breads. To understand the potential chemistry the pH of different fractions/materials and the water contents would have been very useful. Colours of crumb and crust also would be good.

5.    For me the vital points that needs clarification are the units and whether these relate to the wet weight or dry weight of the sample. In line 169 freeze drying of the samples is mentioned, with some generalised results, but I am unsure at which point the samples are freeze dried (I don't think it is mentioned in the methods). Of course the interpretation of the levels of analytes in crust and crumb etc in Figure 1 will be totally different if loss of water has not considered. The legend refers to pmol/mg but not whether this is DRY WEIGHT or WET WEIGHT.  

6.    Just as general comments, it would help the reader to get to grips with things faster if the order Commercial/Homemade was always the in the same order. Why are the axis used for the different analytes different (Fig.1) between the homemade and commercial on different scales? One has to peer at the levels to be able to compare them- and I suppose we are supposed to compare. I can’t read the values on the linking bars that show significant differences- so remove and replace with something readable, without having to zoom.

I would recommend accepting the paper, as long as the wet weight/dry weight issue is clarified. 

Author Response

Please see attached response file

Reviewer 2 Report

Although presented topic seems to be interesting, it is rather a preliminary study and short communications that full research paper. Authors did not determine any antioxidant properties. Conclusion is also very vague. Authors only stated that that low amounts of itaconate isomers occur in bread and related baked goods and it should, further investigations should be performed into their occurrence in other foods. So they should extend their study to other kind of foods.

Thus I did not recommend this paper to publish in such high quality journal  as Antioxidants.  

Author Response

Please see attached response file

Reviewer 3 Report

The work presented in this manuscript is original, interesting and relevant. The study is simple, because the authors only evaluated the content of some compounds in different samples. I missed the study the effect of other factors on the content of the tested compounds or their kinetics of formation. However, the study was properly designed and executed. Sophisticate and adequate equipment was used. The data are well presented and discussed.  I only recommend to avoid the personal style of writing (i.e. avoid the use of "we"). Another comment is that the e-mail address of Frank Pessler is incomplete (“.de”  is missing). The prefix cis should be in italized letters.

Reviewer 4 Report

This manuscript describes an exploratory study of isomers itaconate, mesaconate, and citraconate, compounds having immunomodulatory, anti-oxidative, and antimicrobial properties, in one of the most common staple foods: bread. The authors compared their content in commercial and homemade breads, as well as the leaving agent used for the homemade bread.

The paper is properly written, and the topic is of high interests due to its relatively newness. The approach taken, and methods used to analyze the isomers content look adequate. Nevertheless, there was a clear misunderstanding on the proper terminology for sourdough e sourdough bread. Improvement should be made throughout the text, mostly in the material and methods section and in the discussion.

I suggest to better explain in the introduction what are the main pathways for the production of itaconic acid and its isomers and how processing methods (cooking or fermentation) affect their content.

L62-70: I believe there’s some confusion about the meaning of sourdough. It seems from the way the authors describe it, that they refer to “sourdough” when they talk about the bread dough before cooking, whereas they identify the leavening agent as “sourdough starter”. As abundantly described in the literature and recently review by De Vuyst et al. (Critical Reviews in Food Science and Nutrition, 2021:1-33) sourdough is a mixture of flour and water subjected to fermentation, which can be classified according to the inoculum used to initiate the fermentation or the production process.

Although it appears that what the authors call “sourdough starter” is indeed a type I sourdough, in the methods described in paragraph 2.1 there is no mention of a backslopping procedure. Did the authors used bakery yeast for the first 24h of fermentation? in that case their “sourdough starter” might be a type 0 sourdough. Please clarify this aspect and avoid referring to the bread dough before baking as sourdough, that is objectively incorrect.

L78-79: more information about the commercial breads should be provided. Are those sourdough bread as well? Knowing such information can be of help to better explain the results obtained.

Indeed, the discussion section lack a proper explanation of what determined such different results among homemade and commercial bread, especially for mesaconate and cis-aconitate. Again, I believe the production process, specifically the sourdough fermentation, directly or indirectly, played a great role.

Author Response

Please see attached response file

Round 2

Reviewer 2 Report

The authors improved some parts of the manuscript and I changed my position. However, I suggest extending the conclusions.

Author Response

We extended the Conclusions section by adding a sentence about heat conversion from cis-aconitate and other, unknown, precursors. Unfortunately, we could not add this to the Abstract because we were already over the maximum word count.  

Reviewer 4 Report

Dear Editor,

authors revised the manuscript as suggested. 

It can now be accepted for publication.

Author Response

Thank you for the positive evaluation.